# A Unified Framework for Entropy Search and Expected Improvement in Bayesian Optimization

**Nuojin Cheng** * [1]  **Leonard Papenmeier** * [2]  **Stephen Becker** [1]  **Luigi Nardi** [2] [3]

## Abstract

Bayesian optimization is a widely used method for optimizing expensive black-box functions, with Expected Improvement being one of the most commonly used acquisition functions. In contrast, information-theoretic acquisition functions aim to reduce uncertainty about the function's optimum and are often considered fundamentally distinct from EI. In this work, we challenge this prevailing perspective by introducing a unified theoretical framework, Variational Entropy Search, which reveals that EI and information-theoretic acquisition functions are more closely related than previously recognized. We demonstrate that EI can be interpreted as a variational inference approximation of the popular information-theoretic acquisition function, named Max-value Entropy Search. Building on this insight, we propose VES-Gamma, a novel acquisition function that balances the strengths of EI and MES. Extensive empirical evaluations across both low- and high-dimensional synthetic and real-world benchmarks demonstrate that VES-Gamma is competitive with state-of-the-art acquisition functions and in many cases outperforms EI and MES.

## 1. Introduction

Bayesian optimization (BO) is a widely used technique for maximizing black-box functions. Given a function $f : \mathcal{X} \to \mathbb{R}$, BO iteratively refines a probabilistic surrogate of $f$, typically a Gaussian process (GP), and selects the next evaluation point accordingly. At each iteration, the next sampling point is determined by maximizing an acqui-

sition function (AF) $\alpha : \mathcal{X} \to \mathbb{R}$. An effective AF must balance the exploration-exploitation trade-off, where exploitation prioritizes sampling points predicted by the surrogate to yield high objective values, while exploration targets regions with the potential to uncover even better values.

Expected Improvement (EI) (Mockus, 1998) is one of the most widely used AFs, valued for its simple formulation, computational efficiency, and strong empirical performance. The core idea behind EI is to maximize the expected improvement over the current best observed value, which typically requires a noise-free assumption. More recently, (Villemonteix et al., 2009; Hennig & Schuler, 2012) have introduced the concepts of information-theoretic AFs, which represents a paradigm shift in Bayesian optimization. Unlike EI, which focuses on directly maximizing potential improvement, information-theoretic AFs aim to reduce uncertainty about the function $f$'s optimal position and/or value, often through entropy-based measures. Due to their fundamentally different underlying philosophies and selection criteria, EI and information-theoretic AFs are widely regarded as distinct methodologies within the BO community (Hennig et al., 2022).

Despite their apparent differences, we argue that EI and information-theoretic AFs share deeper theoretical connections than previously recognized. Understanding this relationship is crucial, as it provides novel insights into designing new acquisition functions. By unifying the perspectives of both sides, we introduce VES-Gamma, a new AF that effectively balances their strengths, resulting in a robust AF that adapts well to diverse optimization problems. VES-Gamma inherits the performance of EI while incorporating information-theoretic considerations.

In summary, we make the following key contributions:

1. We introduce the Variational Entropy Search (VES) framework which shows that EI can be interpreted as a special case of the popular information-theoretic acquisition function Max-value Entropy Search (MES). This unified theoretical perspective reveals that these two types of AFs are more closely related than previously recognized.

*Equal contribution  [1]Department of Applied Mathematics, University of Colorado Boulder  [2]Department of Computer Science, Lund University  [3]DBtune. Correspondence to: Nuojin Cheng <Nuojin.Cheng@colorado.edu>.

*Proceedings of the $42^{nd}$ International Conference on Machine Learning*, Vancouver, Canada. PMLR 267, 2025. Copyright 2025 by the author(s).

2. We propose VES-Gamma as an intermediary between EI and MES, incorporating information-theoretic principles while maintaining EI's strength in performance.

3. We provide an extensive evaluation across a diverse set of low- and high-dimensional synthetic, GP samples, and real-world benchmarks, demonstrating that VES-Gamma consistently performs competitively and, in many cases, outperforms both EI and MES.

## 2. Background and Related Work

### 2.1. Gaussian Processes

A Gaussian process is a stochastic process that models an unknown function. It is characterized by the property that any finite set of function evaluations follows a multivariate Gaussian distribution. Assuming that $f$ has a zero mean, a Gaussian process is uniquely determined by the current observations $\mathcal{D}_t \coloneqq \{(\boldsymbol{x}_i, y_{\boldsymbol{x}_i})\}_{i=1}^t$ and the kernel function $\kappa(\boldsymbol{x}, \boldsymbol{x}')$. Given these, at stage $t$, the predicted mean of $y_{\boldsymbol{x}}$ at a new point $\boldsymbol{x}$ is $\mu_t(\boldsymbol{x}) = \boldsymbol{\kappa}_t(\boldsymbol{x})^T (\boldsymbol{K}_t)^{-1} \boldsymbol{y}_t$, and the predicted covariance between points $\boldsymbol{x}$ and $\boldsymbol{x}'$ is $\text{Cov}_t(\boldsymbol{x}, \boldsymbol{x}') = \kappa(\boldsymbol{x}, \boldsymbol{x}') - \boldsymbol{\kappa}_t(\boldsymbol{x})^T (\boldsymbol{K}_t)^{-1} \boldsymbol{\kappa}_t(\boldsymbol{x}')$, where $[\boldsymbol{\kappa}_t(\boldsymbol{x})]_i = \kappa(\boldsymbol{x}_i, \boldsymbol{x})$, $[\boldsymbol{y}_t]_i = y_{\boldsymbol{x}_i}$, and $[\boldsymbol{K}_t]_{i,j} = \kappa(\boldsymbol{x}_i, \boldsymbol{x}_j)$; see Rasmussen et al. (2006) for more details.

### 2.2. Acquisition Functions

Various acquisition functions (AFs) have been proposed to balance exploration and exploitation in optimization tasks, each tailored to different problem characteristics and assumptions. These include Probability of Improvement (PI), Expected Improvement (EI) (Mockus, 1998; Jones et al., 1998), Upper Confidence Bound (UCB) (Srinivas et al., 2010), Knowledge Gradient (KG) (Frazier et al., 2008), and information-theoretic AFs (Villemonteix et al., 2009; Hennig & Schuler, 2012; Hernández-Lobato et al., 2014; Wang & Jegelka, 2017; Hvarfner et al., 2022; Tu et al., 2022). Below, we discuss two types of acquisition functions relevant to this study.

**Expected Improvement.** Expected Improvement (EI) is one of the most commonly used acquisition functions and is formulated as follows:

$$\alpha_{\text{EI}}(\boldsymbol{x}) = \mathbb{E}_{p(y_{\boldsymbol{x}} | \mathcal{D}_t)} \big[ \max\{y_{\boldsymbol{x}}, y_t^*\} \big] - y_t^*, \qquad (1)$$

where $y_t^*$ is the maximum observed value in $\mathcal{D}_t$, and $\mathbb{E}_{p(\cdot)}$ denotes the expectation with respect to the predictive density $p(\cdot)$. The $-y_t^*$ term at the end can be dropped since it is constant with respect to $\boldsymbol{x}$.

**Information-Theoretic AFs.** Information-theoretic AFs form a family of methods designed to select $\boldsymbol{x}$ such

that its evaluation reduces uncertainty regarding the optimal points of the objective function. This uncertainty is quantified using differential entropy, defined as $\mathbb{H}[y] \coloneqq \mathbb{E}_{p(y)}[-\log p(y)]$. Similarly, the conditional entropy is expressed as $\mathbb{H}[y|x] \coloneqq \mathbb{H}[x, y] - \mathbb{H}[x]$.

The first information-theoretic AF for BO is Entropy Search (ES) (Hennig & Schuler, 2012), which is formulated as:

$$\alpha_{\text{ES}}(\boldsymbol{x}) = \mathbb{H}[\boldsymbol{x}^* \mid \mathcal{D}_t] - \mathbb{E}_{p(y_{\boldsymbol{x}} | \mathcal{D}_t)} \left[ \mathbb{H}[\boldsymbol{x}^* \mid \mathcal{D}_t, y_{\boldsymbol{x}}] \right]. \quad (2)$$

Here, the random variable $\boldsymbol{x}^*$ represents the location of the maximum.

Predictive Entropy Search (PES) (Hernández-Lobato et al., 2014) offers a reformulation of ES that is computationally more efficient:

$$\alpha_{\text{PES}}(\boldsymbol{x}) = \mathbb{H}[y_{\boldsymbol{x}} \mid \mathcal{D}_t] - \mathbb{E}_{p(\boldsymbol{x}^* | \mathcal{D}_t)} \left[ \mathbb{H}[y_{\boldsymbol{x}} \mid \mathcal{D}_t, \boldsymbol{x}^*] \right]. \quad (3)$$

Since directly estimating the entropy with $\boldsymbol{x}^*$ is expensive, following the PES format, Max-value Entropy Search (MES) (Wang & Jegelka, 2017) introduced an alternative approach that focuses on reducing the differential entropy of the 1D maximum value $y^*$:

$$\begin{aligned} \alpha_{\text{MES}}(\boldsymbol{x}) &= \mathbb{H}[y^* \mid \mathcal{D}_t] - \mathbb{E}_{p(y_{\boldsymbol{x}} | \mathcal{D}_t)} \left[ \mathbb{H}[y^* \mid \mathcal{D}_t, y_{\boldsymbol{x}}] \right] \\ &= \underbrace{\mathbb{H}[y_{\boldsymbol{x}} \mid \mathcal{D}_t]}_{\text{closed-form}} - \mathbb{E}_{p(y^* | \mathcal{D}_t)} \underbrace{\left[ \mathbb{H}[y_{\boldsymbol{x}} \mid \mathcal{D}_t, y^*] \right]}_{\text{non-closed-form}}. \quad (4) \end{aligned}$$

Unlike MES and its subsequent extensions (Hvarfner et al., 2022; Takeno et al., 2022) which approximate $p(y_{\boldsymbol{x}} \mid \mathcal{D}_t, y^*)$ using a truncated Gaussian, we focus on directly estimating $p(y^* \mid \mathcal{D}_t, y_{\boldsymbol{x}})$ via variational inference.

### 2.3. Related Work

**Variational Inference and Evidence Lower Bound.** Variational Inference (VI) is a widely used technique in Bayesian modeling to approximate intractable posterior distributions (Paisley et al., 2012; Hoffman et al., 2013; Kingma & Welling, 2014). It relies on maximizing the Evidence Lower Bound (ELBO) to approximate the log-likelihood $\log p(\tilde{\boldsymbol{x}})$ in the presence of latent variables $\boldsymbol{z}$. The log-likelihood can be decomposed as follows:

$$\log p(\tilde{\boldsymbol{x}}) \geq \mathbb{E}_{q(\boldsymbol{z})} \left[ \log \left( \frac{p(\tilde{\boldsymbol{x}} \mid \boldsymbol{z}) p(\boldsymbol{z})}{q(\boldsymbol{z})} \right) \right], \qquad (5)$$

where $p(\boldsymbol{z})$ is a fixed prior distribution, and $q(\boldsymbol{z})$ is a variational approximation to the true posterior $p(\boldsymbol{z} \mid \tilde{\boldsymbol{x}})$. The ELBO is formally defined as:

$$\text{ELBO}(p(\tilde{\boldsymbol{x}} \mid \boldsymbol{z}); q(\boldsymbol{z})) \coloneqq \mathbb{E}_{q(\boldsymbol{z})} \left[ \log \left( \frac{p(\tilde{\boldsymbol{x}} \mid \boldsymbol{z}) p(\boldsymbol{z})}{q(\boldsymbol{z})} \right) \right]. \tag{6}$$

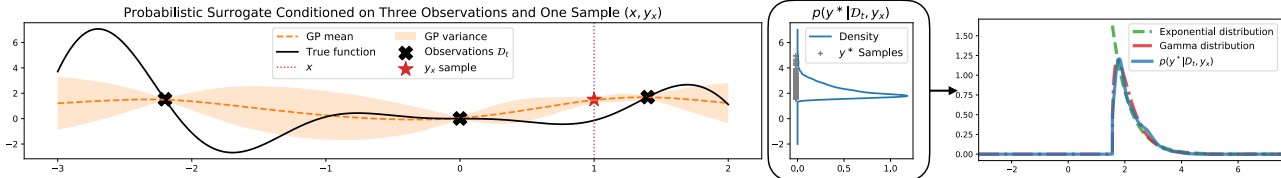

Figure 1. MES aims to optimize $\boldsymbol{x}$ such that the entropy (averaged over all $y_{\boldsymbol{x}}$) of the maximum values $p(y^* \mid \mathcal{D}_t, y_{\boldsymbol{x}})$ is reduced. The left figure illustrates a noiseless Gaussian process conditioned on the observations $\mathcal{D}_t$ with three points (black crosses) and a sample $y_{\boldsymbol{x}}$ at $x = 1$ drawn from $p(y_{\boldsymbol{x}} \mid \mathcal{D}_t)$ (red star). The mid and right panels illustrate the density $p(y^* \mid \mathcal{D}_t, y_{\boldsymbol{x}})$ (blue curves). When $p(y^* \mid \mathcal{D}_t, y_{\boldsymbol{x}})$ is approximated using an exponential distribution (green dashed curve), this leads to the VES-Exp AF that is equivalent to EI. Furthermore, VES-Gamma, which approximates $p(y^* \mid \mathcal{D}_t, y_{\boldsymbol{x}})$ using a Gamma distribution (red dash-dot curve), leads to a more accurate approximation and a generalized version of EI.

By maximizing the ELBO, VI indirectly maximizes the log-likelihood $\log p(\tilde{\boldsymbol{x}})$, thereby improving the quality of the posterior approximation. In many applications, such as variational autoencoders (VAEs) (Kingma & Welling, 2014) and variational diffusion (Kingma et al., 2021), both the conditional likelihood $p(\tilde{\boldsymbol{x}} \mid \boldsymbol{z})$ and the variational distribution $q(\boldsymbol{z})$ are parameterized using neural networks. Since both the expectation reference probability and the term inside the ELBO are parameterized, one common strategy is to estimate the gradient using finite Monte Carlo samples and the *reparameterization trick* to optimize the parameters. We adopt this approach, which enables efficient gradient-based optimization and has been widely applied in the BO community (Wilson et al., 2017).

**Improving the Expected Improvement.** It is widely recognized that EI can be prone to over-exploitation (Qin et al., 2017; Berk et al., 2019; De Ath et al., 2021). To mitigate this issue, Hoffman et al. (2011) and Kandasamy et al. (2020) propose to use a portfolio of AFs, which assigns probabilities to different AFs at each step. Snoek et al. (2012) proposed a fully-Bayesian treatment on EI to improve empirical performance. Another approach is Weighted EI (WEI), which adaptively adjusts the weights of the components within the EI acquisition function (Sóbester et al., 2005; Benjamins et al., 2023). Similarly, Qin et al. (2017) suggest "weakening" EI using suboptimal points suggested by the AF to mitigate its over-exploitative behavior. However, these methods are primarily based on heuristics. Furthermore, information-theoretic acquisition functions are often excluded from these design enhancements, as they are generally considered distinct from heuristic AFs such as PI, EI, UCB, or KG.

**Entropy Approximation in Information-theoretic AFs.** Estimating entropy in information-theoretic acquisition functions is computationally expensive and typically requires approximation techniques. Methods such as ES and PES employ sampling-based approaches, including Markov chain Monte Carlo and expectation propagation. In contrast, MES derives an explicit approximation (Wang & Jegelka, 2017, Eq. 6), which was later interpreted as a variational inference formulation by Takeno et al. (2020). This variational perspective has since been extended to multi-objective optimization (Qing et al., 2023). However, this approximation scheme lacks flexibility in tuning the variational distributions. Furthermore, to the best of our knowledge, most MES-based methods focus on approximating $p(y_{\boldsymbol{x}} \mid y^*, \mathcal{D}_t)$. An exception is Ma et al. (2023), which approximates $p(y^* \mid \mathcal{D}_t, y_{\boldsymbol{x}})$ using a Gaussian distribution. While this approach provides computational advantages, the inherent symmetry of the Gaussian distribution does not align with the properties of $y^*$.

## 3. Variational Entropy Search

### 3.1. Entropy Search Lower Bound

The idea behind our Variational Entropy Search (VES) framework is to maximize a variational lower bound of MES with a predetermined family of densities to approximate $p(y^* \mid \mathcal{D}_t, y_{\boldsymbol{x}})$. Since we assume noiseless observations, the support is $[\max\{y_{\boldsymbol{x}}, y_t^*\}, +\infty)$. VES is illustrated in Figure 1. The lower bound is formalized in Theorem 3.1 and proven in Appendix A.1.

**Theorem 3.1.** *The MES acquisition function in Eq. (4) adheres to the Barber-Agakov (BA) bound (Barber & Agakov, 2004; Poole et al., 2019) and can be bounded from below as follows:*

$$\alpha_{MES}(\boldsymbol{x}) = \mathbb{H}[y^* \mid \mathcal{D}_t] - \mathbb{E}_{p(y_{\boldsymbol{x}} \mid \mathcal{D}_t)} \big[ \mathbb{H}[y^* \mid \mathcal{D}_t, y_{\boldsymbol{x}}] \big]$$
$$\geq \mathbb{H}[y^* \mid \mathcal{D}_t] + \mathbb{E}_{p(y^*, y_{\boldsymbol{x}} \mid \mathcal{D}_t)} \big[ \log q(y^* \mid \mathcal{D}_t, y_{\boldsymbol{x}}) \big],$$
$$(7)$$

*where $q(y^* \mid \mathcal{D}_t, y_{\boldsymbol{x}})$ is any chosen density function that is absolutely continuous with respect to $p(y^* \mid \mathcal{D}_t, y_{\boldsymbol{x}})$.*

Since the first term on the right-hand side of Eq. (7), $\mathbb{H}[y^* \mid \mathcal{D}_t]$, is independent of both $q$ and $\boldsymbol{x}$, we can omit it. This leads us to define the remaining term as the Entropy Search

Lower Bound (ESLBO):

$$\text{ESLBO}(\boldsymbol{x}; q) := \mathbb{E}_{p(y^*, y_{\boldsymbol{x}} | \mathcal{D}_t)}\left[\log q(y^* \mid \mathcal{D}_t, y_{\boldsymbol{x}})\right],$$
(8)

where $p(y^*, y_{\boldsymbol{x}} \mid \mathcal{D}_t)$ represents a joint density, which can be sampled using Gaussian process path sampling (Hernández-Lobato et al., 2014; Wang & Jegelka, 2017).

To optimize $\alpha_{\text{MES}}(\boldsymbol{x})$, we adopt the VI approach (Paisley et al., 2012), indirectly maximizing $\alpha_{\text{MES}}(\boldsymbol{x})$ by instead maximizing ESLBO. To ensure computational feasibility, the VI method constrains the density $q$ to a predefined family $\mathcal{Q}$. When parameterizing $q$ within $\mathcal{Q}$, the problem becomes tractable by solving for $q$ and $\boldsymbol{x}$ iteratively, as detailed in Algorithm 1.

Notably, this procedure, known as expectation maximization (EM), is analogous to maximizing the ELBO in Eq. (5). We conclude our discussion by summarizing the correspondence between ESLBO and ELBO in Table 1.

---

**Algorithm 1** VES Framework

---

**Input:** Observations $\mathcal{D}_t$, variational family $\mathcal{Q}$, number of inner iteration $N$
**Output:** Next sampling location $\boldsymbol{x}_{t+1}$
1: initialize $\boldsymbol{x}_{t+1}^{(0)}$
2: **for** $n = 1 : N$ **do**
3: $\quad q^{(n)}(y^*) \leftarrow \arg\max_{q \in \mathcal{Q}} \text{ESLBO}(\boldsymbol{x}_{t+1}^{(n-1)}; q)$
4: $\quad \boldsymbol{x}_{t+1}^{(n)} \leftarrow \arg\max_{\boldsymbol{x}_{t+1}} \text{ESLBO}(\boldsymbol{x}_{t+1}; q^{(n)})$
5: **end for**
6: return $\boldsymbol{x}_{t+1}^{(N)}$

---

### 3.2. EI Through the Lens of the VES Framework

In this section, we aim to establish an explicit connection between the VES and EI acquisition functions, allowing us to see EI through the lens of a VI approximation of the information-theoretical MES AF. We define $\mathcal{Q}$ as the set of all exponential density functions, $\mathcal{Q}_{\text{exp}}$, parameterized by the $\lambda > 0$ exponential density parameter and with support bounded from below by $\max\{y_{\boldsymbol{x}}, y_t^*\}$. The variational density function $q$ is given by

$$q(y^* | \mathcal{D}_t, y_{\boldsymbol{x}}; \lambda) = \lambda e^{-\lambda(y^* - \max\{y_{\boldsymbol{x}}, y_t^*\})} \mathbf{1}_{y^* \geq \max\{y_{\boldsymbol{x}}, y_t^*\}}.$$
(9)

For noiseless observations, the indicator function $\mathbf{1}_{y^* \geq \max\{y_{\boldsymbol{x}}, y_t^*\}}$ always equals one and can be omitted. Plugging in $q$ from Eq. (9) into the ESLBO (Eq. (8)) yields a new $\lambda$-parameterized AF. Since this AF stems from the exponential distribution, we name it VES-Exp. Theorem 3.2 shows that the next sampling point generated from VES-Exp within Algorithm 1 will be the same as for the EI AF; the theorem is proven in Appendix A.2.

**Theorem 3.2.** *When the family $\mathcal{Q}_{\text{exp}}$ is selected as in Eq. (9) and the function is noiseless, ESLBO in Eq. (8) turns into*

$$ESLBO(\boldsymbol{x}; \lambda) = \log \lambda - \lambda \underbrace{\mathbb{E}_{p(y^* | \mathcal{D}_t)}[y^*]}_{\text{constant}}$$
$$+ \lambda \underbrace{\mathbb{E}_{p(y_{\boldsymbol{x}} | \mathcal{D}_t)}[\max\{y_{\boldsymbol{x}}, y_t^*\}]}_{\text{EI}}.$$
(10)

*Maximizing $ESLBO(\boldsymbol{x}; \lambda)$ in Eq. (10) with respect to $\boldsymbol{x}$ and $\lambda$ yields the same $\boldsymbol{x}$ solution as the maximization of EI in Eq. (1).*

The key idea behind the proof is that, following Algorithm 1, the ESLBO in Eq. (10) always converges within two iterations. Regardless of the positive value of $\lambda$, the value of $\boldsymbol{x}$ that maximizes $\text{ESLBO}(\boldsymbol{x}; \lambda)$ remains the same. Consequently, starting from an arbitrary initial point $\boldsymbol{x}^{(0)}$, a positive $\lambda^{(1)}$ is derived, ensuring that ESLBO reaches its maximum value in the next iteration.

Theorem 3.2 reveals that EI can be viewed as a special case of MES, giving a new information-theoretic interpretation of the most popular acquisition function in use today. However, the exponential distribution has a fairly rigid parametric form that does not capture the characteristics of $p(y^* \mid \mathcal{D}_t, y_{\boldsymbol{x}})$. Figure 1 (right) shows an example of the structural limitations of the exponential density in green. We generate 1000 samples from an example distribution $p(y^* \mid \mathcal{D}_t, y_{\boldsymbol{x}})$, and observe that it significantly deviates from an exponential distribution. Specifically, the density of $p(y^* \mid \mathcal{D}_t, y_{\boldsymbol{x}})$ is non-monotonic, exhibiting a peak before decreasing near $\max\{y_{\boldsymbol{x}}, y_t^*\}$ (approximately 1.55), while exponential distributions are necessarily monotonic.

This observation motivates the need to enrich the variational distributions $\mathcal{Q}$ to allow more flexibility. A natural extension is to use a Gamma distribution, which is a generalization of the exponential distribution. The Gamma density approximation in the previous example is shown in red in Figure 1 (right). The next section introduces VES-Gamma, which is a more general AF that extends VES-Exp and its equivalent EI acquisition function.

### 3.3. VES-Gamma: A Generalization of EI

VES-Gamma defines $\mathcal{Q}$ as the Gamma distribution parameterized by $k, \beta > 0$ with its support bounded from below by $\max\{y_{\boldsymbol{x}}, y_t^*\}$. The variational density is

$$q(y^* \mid \mathcal{D}_t, y_{\boldsymbol{x}}; k, \beta) = \frac{\beta^k}{\Gamma(k)} (y^* - \max\{y_{\boldsymbol{x}}, y_t^*\})^{k-1}$$
$$\times e^{-\beta(y^* - \max\{y_{\boldsymbol{x}}, y_t^*\})} \mathbf{1}_{y^* \geq \max\{y_{\boldsymbol{x}}, y_t^*\}},$$
(11)

where $\Gamma(\cdot)$ denotes the Gamma function. The noise-free assumption allows us to omit the indicator function, and

Table 1. Comparison of key aspects between the ELBO and ESLBO approaches.

| Property | ELBO Approach | ESLBO Approach |
|---|---|---|
| Primary Variable | $p(\tilde{\boldsymbol{x}} \mid \boldsymbol{z})$ | $\boldsymbol{x}$ |
| Variational Variable | $q(\boldsymbol{z})$ | $q(y^* \mid y_{\boldsymbol{x}}, \mathcal{D}_t)$ |
| Lower Bound Formulation | $\mathrm{ELBO}(q(\boldsymbol{z}); p(\tilde{\boldsymbol{x}} \mid \boldsymbol{z}))$ | $\mathrm{ESLBO}(q; \boldsymbol{x})$ |

the ESLBO is reformulated as

$$
\begin{aligned}
\mathrm{ESLBO}(\boldsymbol{x}; k, \beta) =\ & k \log \beta - \log \Gamma(k) \\
& + (k-1)\mathbb{E}_{p(y^*, y_{\boldsymbol{x}}|\mathcal{D}_t)} \left[\log\left(y^* - \max\{y_{\boldsymbol{x}}, y_t^*\}\right)\right] \\
& - \beta\mathbb{E}_{p(y^*|\mathcal{D}_t)}[y^*] + \beta \underbrace{\mathbb{E}_{p(y_{\boldsymbol{x}}|\mathcal{D}_t)}[\max\{y_{\boldsymbol{x}}, y_t^*\}]}_{\text{EI}}.
\end{aligned}
\tag{12}
$$

The ESLBO in Eq. (12) serves as the primary objective in the VES-Gamma algorithm. Eq. (12) consists of five terms, with the last term being the EI AF in Eq. (1) scaled by a multiplicative factor. The two hyperparameters, $k$ and $\beta$, originally part of the Gamma distribution, dynamically balance different components of the objective. In particular, when $k = 1$, the Gamma distribution reduces to an exponential distribution, making the ESLBO in Eq. (12) equivalent to Eq. (10). In the following section, we discuss the approach for determining values for $k$ and $\beta$.

**Auto-determination of Tradeoff Hyperparameters.** For any fixed $\boldsymbol{x}$, the global maximum of the ESLBO in Eq. (12) with respect to $k$ and $\beta$ uniquely exists, as can be demonstrated through derivative analysis. Taking the partial derivatives of ESLBO in Eq. (12) and setting them to zero, we obtain:

$$
\log \beta - \frac{\partial \log \Gamma(k)}{\partial k} + \mathbb{E}\left[\log z_{\boldsymbol{x}}^*\right] = 0, \quad \frac{k}{\beta} - \mathbb{E}\left[z_{\boldsymbol{x}}^*\right] = 0,
$$

where the random variable $z_{\boldsymbol{x}}^* := y^* - \max\{y_{\boldsymbol{x}}, y_t^*\}$.

Substituting the second equation into the first yields:

$$
\log k - \psi(k) = \log \mathbb{E}[z_{\boldsymbol{x}}^*] - \mathbb{E}[\log z_{\boldsymbol{x}}^*],
\tag{13}
$$

where $\psi(k) := \partial \log \Gamma(k)/\partial k$ is the digamma function (Abramowitz et al., 1988), which can be efficiently approximated as a series. By Jensen's inequality, $\log \mathbb{E}[z_{\boldsymbol{x}}^*] - \mathbb{E}[\log z_{\boldsymbol{x}}^*] \geq 0$. Since $\log k - \psi(k)$ is strictly decreasing and approaches zero asymptotically (see Figure 2), the root of Eq. (13), $k_{\boldsymbol{x}}^*$, exists uniquely—except in the degenerate case where $z_{\boldsymbol{x}}^*$ is deterministic. In the practical implementation, we apply a clamping function to ensure that the term $\log k - \psi(k)$ does not become zero, and we employ a regularization mechanism to keep the resulting root $k_{\boldsymbol{x}}^*$ close to 1. Specifically, we use L2 regularization when solving $\log k - \psi(k) = \log \mathbb{E}[z_{\boldsymbol{x}}^*] - \mathbb{E}[\log z_{\boldsymbol{x}}^*]$ since the unregularized version is unstable, presumably due to a widely flat landscape. In particular, for $\xi(k) :=$

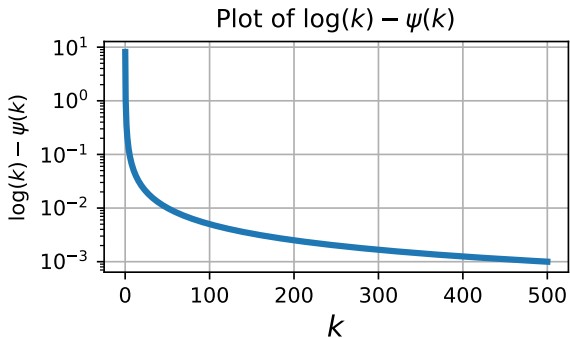

Figure 2. Plot of $\log k - \psi(k)$ for $k \in [0, 500]$. The function is strictly decreasing and asymptotically approaches zero.

$\log k - \psi(k) - \log \mathbb{E}[z_{\boldsymbol{x}}^*] + \mathbb{E}[\log z_{\boldsymbol{x}}^*]$, we solve the following optimization problem:

$$
\min_k \xi(k)^2 + \lambda \left(k - 1\right)^2,
\tag{14}
$$

where $\lambda$ is a regularization parameter which is set to 1 in our experiments.

With this analysis, the value $k_{\boldsymbol{x}}^*$ is determined by minimizing Eq. (14) using Brent's method (Brent, 2013), where expectations of $z_{\boldsymbol{x}}^*$ are estimated via Monte Carlo sampling from $p(y^*, y_{\boldsymbol{x}} \mid \mathcal{D}_t)$. Once $k_{\boldsymbol{x}}^*$ is obtained, the corresponding $\beta_{\boldsymbol{x}}^*$ follows as:

$$
\beta_{\boldsymbol{x}}^* \leftarrow \frac{k_{\boldsymbol{x}}^*}{\mathbb{E}\left[z_{\boldsymbol{x}}^*\right]}.
\tag{15}
$$

Notably, the weighting parameters $k_{\boldsymbol{x}}^*$ and $\beta_{\boldsymbol{x}}^*$ are location-dependent, as $z_{\boldsymbol{x}}^*$ itself varies with $\boldsymbol{x}$. The VES-Gamma algorithm, which incorporates these principles, is detailed in Algorithm 2.

Although we provide both a theoretical justification and a practical implementation for the VES-Gamma AF, a deeper interpretation of the ESLBO in Eq. (12) remains an open research question. Due to the complex non-linear structure of Eq. (12), it is currently uncertain if there is a clear and straightforward interpretation of the various terms and the overall expression. As an example, we hypothesize that the third term acts as an "anti-EI" component, steering the VES-Gamma solution away from the EI recommendation to promote diversity, with the values of $\beta_{\boldsymbol{x}}^*$ and $k_{\boldsymbol{x}}^*$ dynamically balancing its influence. Investigating this hypothesis

**Algorithm 2** VES-Gamma

---

**Input:** Sample set $\mathcal{D}_t$, number of inner iterations $N$
**Output:** Next sampling location $\boldsymbol{x}_{t+1}$

1: initialize $\boldsymbol{x}_{t+1}^{(0)}$
2: **for** $n = 1 : N$ **do**
3:     Evaluate values of $\mathbb{E}\left[z_{\boldsymbol{x}}^*\right]$ and $\mathbb{E}\left[\log\left(z_{\boldsymbol{x}}^*\right)\right]$ by sampling $p(y^*, y_{\boldsymbol{x}} \mid \mathcal{D}_t)$ given $\boldsymbol{x} = \boldsymbol{x}_{t+1}^{(n-1)}$
4:     Solve $k^{(n)}$ from Eq. (13)
5:     Solve $\beta^{(n)}$ from Eq. (15)
6:     Update $\boldsymbol{x}_{t+1}^{(n)} \leftarrow \arg\max_{\boldsymbol{x}} \text{ESLBO}(\boldsymbol{x}; k^{(n)}, \beta^{(n)})$ defined in Eq. (12)
7: **end for**
8: return $\boldsymbol{x}_{t+1}^{(N)}$

---

*Table 2.* Average duration of a BO loop for each AF. We measure the runtime on the `Branin`, `Levy`, and `Hartmann` benchmarks and average over benchmarks, BO iterations, and 10 random restarts. For $N = 5$ outer repetitions, VES has a higher runtime than the other acquisition functions.

| AF | average time per BO iteration |
|-----|-------------------------------|
| EI | $1.627s\,(\pm 0.916s)$ |
| MES | $1.120s\,(\pm 0.472s)$ |
| VES | $10.910s\,(\pm 12.323)$ |

and further elucidating the role of each term within ESLBO will be the focus of future research.

**Computational Cost of VES-Gamma.** Implementing VES-Gamma in Algorithm 2 is computationally intensive. The number of inner iterations, $N$, must be sufficiently large for convergence, and each inner iteration requires estimating $\mathbb{E}[z_{\boldsymbol{x}}^*]$ by sampling a large number of $y^*$. Consequently, the overall BO loop takes significantly more time than EI and MES, as shown in Table 2 for $N = 5$. However, since black-box function evaluations are often expensive, the additional computational cost of VES-Gamma is not a major bottleneck in many real-world applications.

## 4. Results

### 4.1. Experimental Setup

We employ a consistent Gaussian Process (GP) hyperparameter and prior setting across all benchmarks and acquisition functions, evaluating Bayesian optimization (BO) performance using the simple regret $r(t) := f^* - \max_{(\boldsymbol{x}_i, y_{\boldsymbol{x}_i}) \in \mathcal{D}_t} y_{\boldsymbol{x}_i}$, where $f^* := \max_{\boldsymbol{x} \in \mathcal{X}} f(\boldsymbol{x})$. When $f^*$ is unknown, we instead report the negative best function value, $-\max_{(\boldsymbol{x}_i, y_{\boldsymbol{x}_i}) \in \mathcal{D}_t} y_{\boldsymbol{x}_i}$.

To warm-start the optimization process, we initialize with 20 random samples drawn uniformly from $\mathcal{X}$ and model

the GP using a $5/2$-Matérn kernel with automatic relevance determination (ARD) and a dimensionality-scaled length-scale prior (Hvarfner et al., 2024). Following the theoretical assumption in the VES framework, we only focus on experiments with noise-free observations. Although all benchmarks are noiseless, we allow the GP to accommodate potential non-stationarity or discontinuities in the underlying function.

Each experiment is repeated 10 times to estimate average performance, with results reported as mean $\pm$ one standard deviation. For problems with dimension less than 50, we run 100 iterations, otherwise 1000 iterations are computed. For numerical stability in VES-Gamma, we apply clamping: $z_{\boldsymbol{x}}^* = \max\{10^{-10}, y^* - \max\{y_{\boldsymbol{x}}, y_t^*\}\}$. The expectation in Eq. (12) is estimated via pathwise conditioning (Wilson et al., 2021) using 128 posterior samples. Additionally, the number of inner iterations $N$ in Algorithm 2 is set to 5, with early stopping applied if $\|\boldsymbol{x}^{(n-1)} - \boldsymbol{x}^{(n)}\| < d \cdot 10^{-5}$, where $d$ denotes the problem dimension. We implement VES-Gamma and our other experiments using `BoTorch` (Balandat et al., 2020). We always compare against `LogEI` (Ament et al., 2023) and use EI and `LogEI` interchangeably. The code is available in https://github.com/NUOJIN/variational-entropy-search.

**Benchmarks.** To evaluate VES, we consider three distinct categories of benchmark problems: synthetic benchmarks, GP samples, and real-world optimization tasks.

For synthetic benchmarks, we examine commonly used functions that are diverse in dimensionality and landscape complexity. Specifically, we evaluate the 2-dimensional `Branin`, the 4-dimensional `Levy`, the 6-dimensional `Hartmann`, and the 8-dimensional `Griewank` functions. These benchmarks are widely utilized in optimization studies and provide controlled testbeds for algorithmic comparisons (Surjanovic & Bingham).

For GP sample benchmarks, we draw from a GP prior with a $\nu = 5/2$ Matérn kernel. These experiments examine the impact of varying length scales ($\ell = \{0.5, 1, 2\}$) and dimensionalities ($d = \{2, 50, 100\}$) on algorithmic performance.

For real-world scenarios, we utilize a set of benchmarks reflecting practical high-dimensional problems. These include the 60-dimensional `Rover` problem (Wang et al., 2018), the 124-dimensional soft-constrained `Mopta08` (Jones, 2008) benchmark introduced in Eriksson & Jankowiak (2021), the 180-dimensional `Lasso-DNA` problem from `LassoBench` (Šehić et al., 2022), and the 388-dimensional `SVM` benchmark, also introduced in Eriksson & Jankowiak (2021). These tasks represent optimization challenges in engineering design, machine learning, and computational biology.

Due to space constraints, additional experiments are provided in Appendix B.

## 4.2. Comparing VES-Exp and EI

**Kolmogorov-Smirnov Test.** After establishing the theoretical equivalence of VES-Exp and EI in Section 3.2, we aim to validate this equivalence in our practical implementation. To this end, we employ the Kolmogorov-Smirnov (KS) two-sample test with a significance level of $\alpha = 5\%$ to assess statistical similarity. The two samples consist of function values evaluated by each acquisition function (AF) across 10 repeated trials, i.e., $Y_{\mathrm{EI}}(t) := \{\boldsymbol{y}_t^i\}_{i=1}^{10}$, where $\boldsymbol{y}_t^i$ denotes the function evaluation at step $t$ in the $i$-th trial. The null hypothesis states that the function evaluations from VES-Exp and EI originate from the same distribution.

We collect function values for all $500$ iterations and consider a test successful (pass) for each iteration $t$ if the null hypothesis is not rejected. We include six different benchmarks spanning low-dimensional synthetic problems to high-dimensional real-world scenarios. Additional implementation details on KS test are presented in Appendix C.

**Empirical Equivalence Results** Figure 3 illustrates the function values obtained by VES-Exp and EI, while Table 3 reports the passing rates of the KS test across six benchmarks. The results show that all passing rates exceed $90\%$, with the Hartmann benchmark achieving the highest proportion of accepted tests.

Several factors explain the remaining discrepancies between VES-Exp and EI. First, since both acquisition functions are non-convex, their optimization may yield different next sampling points $\boldsymbol{x}_{t+1}$ due to variations in initialization. Second, VES methods employ a clamping mechanism to ensure that $z_{\boldsymbol{x}}^*$ remains numerically positive, which introduces a dependency between $y^*$ and $\boldsymbol{x}$. In practice, this violates the assumptions used in the proof in Appendix A.2. We also employed Log-EI (Ament et al., 2023) instead of EI in our experiment, which may also explain the difference. Finally, while EI has a closed-form expression, VES-Exp relies on Monte Carlo estimation, introducing numerical inexactness and potential discrepancies.

## 4.3. Performance of VES-Gamma

**Synthetic Test Functions.** Figure 4 illustrates the performance of various methods, including MES, EI, and VES-Gamma, across four synthetic benchmark functions: Branin ($d = 2$), Levy ($d = 4$), Hartmann ($d = 6$), and Griewank ($d = 8$). The metric shown is the logarithm of the best value (or simple regret), averaged over 10 independent runs.

*Table 3.* Kolmogorov-Smirnov two-sample test passing rate between VES-Exp and EI for various benchmarks. More details about p-values are available in Figure 8 in the appendix.

|  | Passing Rate (%) |
|---|---|
| Branin ($d = 2$) | 94.00 |
| Hartmann ($d = 6$) | 99.80 |
| Rover ($d = 60$) | 92.60 |
| Mopta08 ($d = 124$) | 93.20 |
| Prior ($d = 2$) | 93.60 |
| Prior ($d = 50$) | 94.60 |

On Branin, VES-Gamma achieves the best performance, with MES and EI lagging behind. For Levy, VES-Gamma and EI are effectively tied for the best results, with MES showing slightly worse performance. On the Hartmann function, VES-Gamma outperforms all other methods. Finally, for the Griewank function, VES-Gamma and EI once again demonstrate similar performance, significantly outperforming MES.

Overall, these results highlight the robustness of VES-Gamma across diverse synthetic benchmarks, consistently ranking among the top-performing methods.

**GP Samples.** Here, we study problem instances where the GP can be fitted without model mismatch. To this end, we sample realizations from an isotropic 100-dimensional GP prior with varying length scale $\ell = 0.05, 0.1, 0.25, 0.5$, using the same $5/2$-Matérn covariance function for the GP prior and the GP we fit to the observations.

Figure 5 shows the optimization performance on the 100-dimensional GP prior samples. For $\ell = 0.05, 0.1, 0.25$, VES-Gamma outperforms EI and MES by a wide margin. EI and MES converge to a suboptimal solution. Only for $\ell = 0.5$ does EI reach the same quality as VES-Gamma, outperforming MES.

**Real-World Benchmarks.** Figure 6 presents the performance of VES-Gamma, EI, and MES across four real-world optimization problems: the 60-dimensional Rover trajectory optimization, the 124-dimensional Mopta08 vehicle optimization, the 180-dimensional weighted Lasso-DNA regression, and the 388-dimensional SVM hyperparameter tuning benchmarks.

Consistent with previous observations, VES-Gamma delivers strong performance, significantly outperforming all other acquisition functions on the SVM benchmark. It also ranks among the top-performing methods, alongside EI, on the Mopta08 and Lasso-DNA benchmarks. On the Rover problem, VES-Gamma performs comparably to EI, while MES achieves the best results in this scenario.

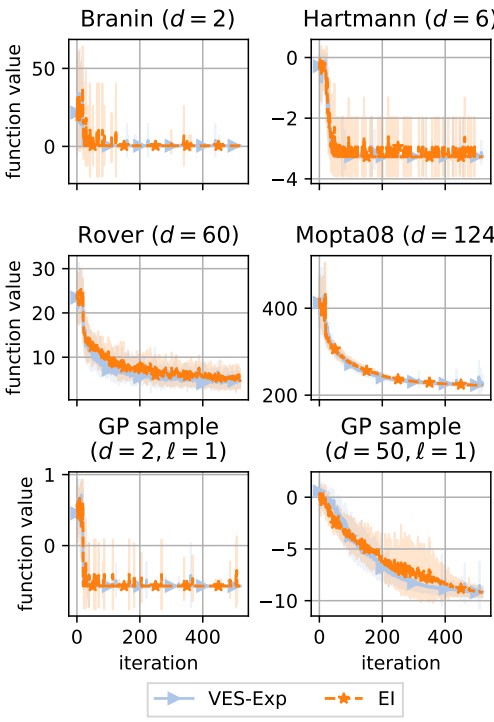

*Figure 3.* Function values observed at each BO iteration for the EI and VES-Exp acquisition functions.

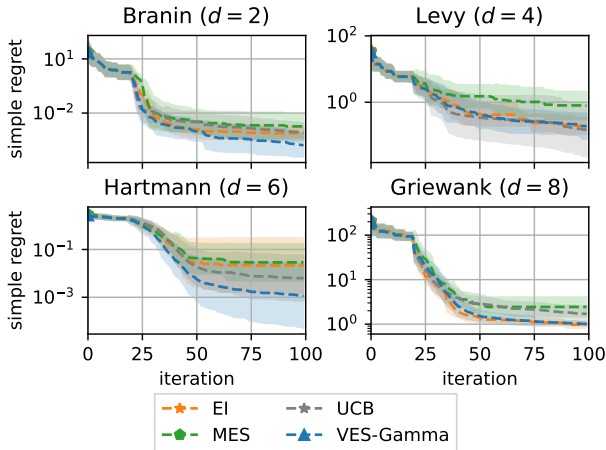

*Figure 4.* VES-Gamma, EI, and MES on the synthetic `Branin` ($d = 2$), `Levy` ($d = 4$), `Hartmann` ($d = 6$), and `Griewank` ($d = 8$) benchmark functions. Average log simple regret: VES-Gamma performs best on `Branin` and `Hartmann`, and it is competitive on `Levy` and `Griewank`.

MES exhibits mixed performance across the benchmarks, achieving the best results on `Rover` but falling behind on the `Mopta08` and `SVM` problems.

Overall, VES-Gamma demonstrates robust and consistent performance across all benchmarks, establishing itself as a versatile and reliable acquisition function for high-dimensional real-world optimization problems.

## 5. Conclusion

In this work, we introduce Variational Entropy Search (VES), a unified framework that bridges Expected Improvement (EI) and information-theoretic acquisition functions through a variational inference approach. We demonstrate that EI can be interpreted as a special case of Max-value Entropy Search (MES), revealing a deeper theoretical connection between these two widely used methodologies in Bayesian optimization. Building on this insight, we propose VES-Gamma, a novel acquisition function that dynamically balances the strengths of EI and MES. Comprehensive benchmark evaluations across a diverse set of low- and high-dimensional optimization problems highlight the robust and consistently high performance of VES-Gamma. These results underscore the potential of the VES framework as a promising foundation for developing more adaptive and efficient acquisition functions in Bayesian optimization.

**Limitations and future work.** While the Gamma distribution offers flexibility, future work will explore alternative variational distributions to enhance the adaptability of VES-Gamma. Another key direction is improving computational efficiency. Additionally, extending the theoretical framework to noisy settings remains an open challenge, requiring adaptations in variational inference to account for stochastic density supports.

## Acknowledgements

This project was partly supported by the Wallenberg AI, Autonomous Systems, and Software program (WASP) funded by the Knut and Alice Wallenberg Foundation, the AFOSR awards FA9550-20-1-0138, with Dr. Fariba Fahroo as the program manager, DOE award DE-SC0023346, and by the US Department of Energy's Wind Energy Technologies Office. The computations were enabled by resources provided by the National Academic Infrastructure for Supercomputing in Sweden (NAISS), partially funded by the Swedish Research Council through grant agreement no. 2022-06725

## Impact Statement

This paper presents work that aims to advance the field of Machine Learning. There are many potential societal consequences of our work, none of which we feel must be specifically highlighted here.

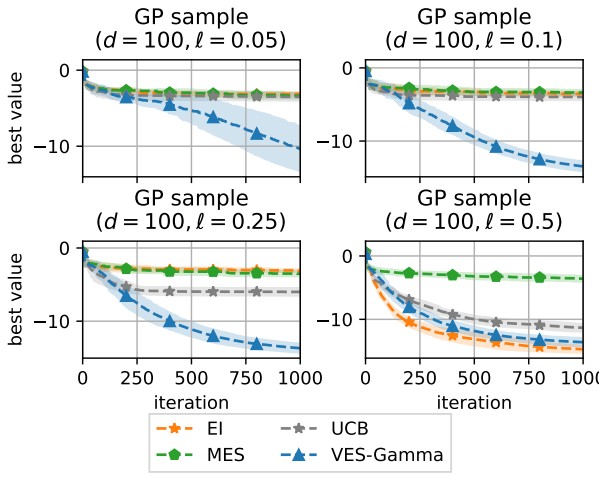

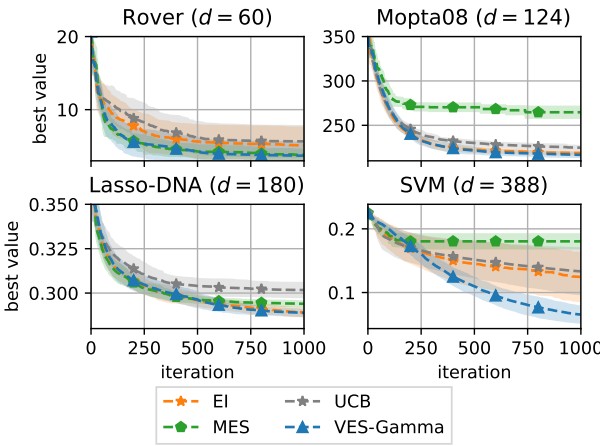

*Figure 5.* Performance curves (best values up to each iteration). VES-Gamma shows superior performance on all but one problem where it performs as good as EI.

*Figure 6.* Performance curves (best function value up to each iteration). VES-Gamma outperforms all other AFs on `SVM` and performs well on the other problems.

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

# A. Proofs

## A.1. ESLB Proof

The MES acquisition function in Eq. (4) can be lower bounded as follows,

*Proof.*

$$
\begin{aligned}
\alpha_{\mathrm{MES}}(\boldsymbol{x}) &= \mathbb{H}[y^* \mid \mathcal{D}_t] - \mathbb{E}_{p(y_{\boldsymbol{x}}|\mathcal{D}_t)}\mathbb{H}[y^* \mid \mathcal{D}_t, y_{\boldsymbol{x}}] \\
&= \mathbb{H}[y^* \mid \mathcal{D}_t] + \mathbb{E}_{p(y^*,y_{\boldsymbol{x}}|\mathcal{D}_t)}[\log(p(y^* \mid \mathcal{D}_t, y_{\boldsymbol{x}}))] \\
&= \mathbb{H}[y^* \mid \mathcal{D}_t] + \mathbb{E}_{p(y^*,y_{\boldsymbol{x}}|\mathcal{D}_t)}\left[\log\left(\frac{p(y^* \mid \mathcal{D}_t, y_{\boldsymbol{x}})q(y^* \mid \mathcal{D}_t, y_{\boldsymbol{x}})}{q(y^* \mid \mathcal{D}_t, y_{\boldsymbol{x}})}\right)\right] \\
&= \mathbb{H}[y^* \mid \mathcal{D}_t] + \mathbb{E}_{p(y^*,y_{\boldsymbol{x}}|\mathcal{D}_t)}[\log(q(y^* \mid \mathcal{D}_t, y_{\boldsymbol{x}}))] + \mathbb{E}_{p(y_{\boldsymbol{x}}|\mathcal{D}_t)}[D_{\mathrm{KL}}(p(y^* \mid \mathcal{D}_t, y_{\boldsymbol{x}})\|q(y^* \mid \mathcal{D}_t, y_{\boldsymbol{x}}))] \\
&\geq \mathbb{H}[y^* \mid \mathcal{D}_t] + \mathbb{E}_{p(y^*,y_{\boldsymbol{x}}|\mathcal{D}_t)}[\log(q(y^* \mid \mathcal{D}_t, y_{\boldsymbol{x}}))],
\end{aligned}
$$

where the KL divergence $D_{\mathrm{KL}}(p(x)\|q(x)) := \mathbb{E}_{p(x)}[\log(p(x)/q(x))]$. The inequality is tight if and only if $\mathbb{E}_{p(y_{\boldsymbol{x}}|\mathcal{D}_t)}[D_{\mathrm{KL}}(p(y^* \mid \mathcal{D}_t, y_{\boldsymbol{x}})\|q(y^* \mid \mathcal{D}_t, y_{\boldsymbol{x}}))] = 0$, which implies $p(y^* \mid \mathcal{D}_t, y_{\boldsymbol{x}}) = q(y^* \mid \mathcal{D}_t, y_{\boldsymbol{x}})$ for all $y_{\boldsymbol{x}} \mid \mathcal{D}_t$. $\square$

## A.2. VES-Exp and EI Algorithmic Equivalence

Theorem 3.2 is proved as follows:

*Proof.* By restricting the variational distributions to exponential distributions, we slightly abuse the input notations of ESLBO in (8) and define:

$$
\begin{aligned}
\mathrm{ESLBO}(\lambda, \boldsymbol{x}) &= \mathbb{E}_{p(y^*,y_{\boldsymbol{x}}|\mathcal{D}_t)}[\log(\lambda \exp(-\lambda(y^* - \max\{y_{\boldsymbol{x}}, y_t^*\})))] \\
&= \log \lambda - \lambda \mathbb{E}_{p(y^*,y_{\boldsymbol{x}}|\mathcal{D}_t)}[(y^* - \max\{y_{\boldsymbol{x}}, y_t^*\})] \\
&= \log \lambda - \lambda \underbrace{\mathbb{E}_{p(y^*|\mathcal{D}_t)}[y^*]}_{\text{constant}} + \lambda \underbrace{\mathbb{E}_{p(y_{\boldsymbol{x}}|\mathcal{D}_t)}[\max\{y_{\boldsymbol{x}}, y_t^*\}]}_{\text{EI AF}}.
\end{aligned}
\tag{16}
$$

Beginning with an arbitrary initial value $\boldsymbol{x}^{(0)}$, we determine the corresponding parameter

$$
\lambda^{(1)} = \frac{1}{\mathbb{E}_{p(y^*,y_{\boldsymbol{x}^{(0)}}|\mathcal{D}_t)}[(y^* - \max\{y_{\boldsymbol{x}^{(0)}}, y_t^*\})]},
\tag{17}
$$

which is derived by taking the derivative of (16) and letting it equal zero. With $\lambda$ fixed, $\mathrm{ESLBO}(\lambda^{(1)}, \boldsymbol{x})$ produces the same result as the EI acquisition function in (1). We then compute $\lambda^{(2)}$ based on $\boldsymbol{x}^{(1)}$ following (17). Regardless of the specific value of $\lambda^{(2)}$, the ESLBO function consistently yields the same result, $\boldsymbol{x}^{(1)}$. This consistency ensures that the VES iteration process converges in a single step. The final outcome, represented as $(\boldsymbol{x}^{(1)}, \lambda^{(2)})$, indicates that the corresponding $q(y^* \mid y_{\boldsymbol{x}}, \mathcal{D}_t)$ is the closest approximation to $p(y^* \mid y_{\boldsymbol{x}}, \mathcal{D}_t)$ within $\mathcal{Q}_{\mathrm{exp}}$ (in the sense that minimizes their KL divergence). $\square$

# B. Additional Experimental Results

In this section, we evaluate VES-Gamma (Algorithm 2) on additional benchmarks.

### B.1. Synthetic Test Functions

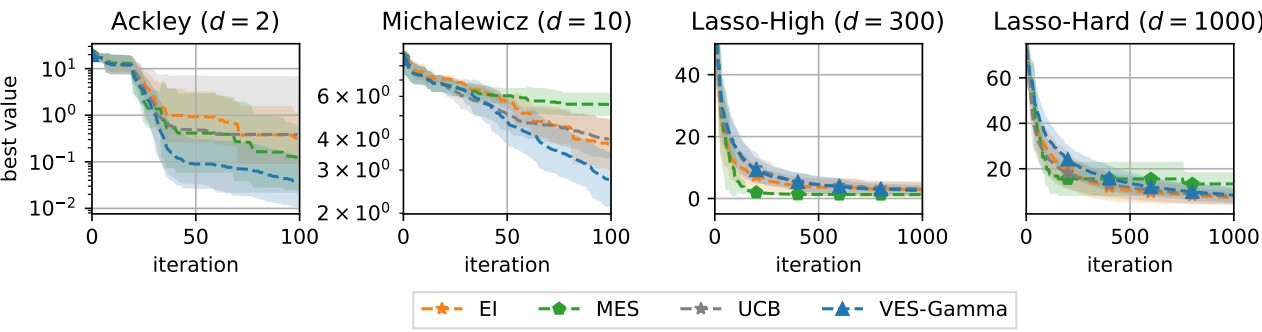

*Figure 7.* Performance plots for EI, MES, and VES-Gamma on additional synthetic benchmark functions. VES-Gamma shows robust performance throughout the bank.

Figure 7 shows the performance of the different acquisition functions, EI, MES, and VES-Gamma, on additional synthetic benchmark functions: the `Ackley` and `Michalewicz` test functions[1] and the `Lasso-High` and `Lasso-Hard` benchmarks (Šehić et al., 2022). On the 1000-dimensional `Lasso-Hard` problem, VES-Gamma ran into a timeout after 48 hours. Therefore, we plot the mean up to the minimum number of iterations performed across all repetitions. VES-Gamma demonstrates robust performance across the benchmarks, outperforming all other acquisition functions on `Ackley`, MES on `Michalewicz`, and performing similarly to the other acquisition functions on the `Lasso` benchmarks. VES-Gamma and MES perform considerably worse than VES-Gamma, especially on the more high-dimensional problems.

## C. Kolmogorov-Smirnov Test Statistic

The Kolmogorov-Smirnov (KS) two-sample test is a non-parametric statistical method used to determine whether two samples are drawn from the same continuous distribution. It compares their empirical cumulative distribution functions (ECDFs) and calculates a test statistic that quantifies their maximum difference. Given two independent samples as function evaluations from VES-Exp $\{X_1, X_2, \ldots, X_{n_1}\}$ and from EI $\{Y_1, Y_2, \ldots, Y_{n_2}\}$, their ECDFs are defined as:

$$F_X(x) = \frac{1}{n_1} \sum_{i=1}^{n_1} \mathbb{I}(X_i \leq x), \quad F_Y(x) = \frac{1}{n_2} \sum_{j=1}^{n_2} \mathbb{I}(Y_j \leq x),$$

where $\mathbb{I}(\cdot)$ is the indicator function, equal to 1 if the condition is true and 0 otherwise. The KS test statistic is given by:

$$D = \sup_x |F_X(x) - F_Y(x)|,$$

where $\sup_x$ denotes the supremum over all possible values of $x$. This statistic measures the maximum absolute difference between the ECDFs of the two samples.

**Statistical Hypotheses.** The hypotheses for the KS test are defined as:

- Null hypothesis ($H_0$): $F_X(x) = F_Y(x)$ for all $x$ (the two samples come from the same distribution).

- Alternative hypothesis ($H_a$): $F_X(x) \neq F_Y(x)$ for at least one $x$ (the two samples come from different distributions).

To test these hypotheses, the test p-value is solved using the Kolmogorov-Smirnov survival function:

$$p_{\text{test}} = Q_{\text{KS}} \left( \sqrt{\frac{n_1 n_2}{n_1 + n_2}} D \right),$$

---

[1] `https://www.sfu.ca/~ssurjano/optimization.html`

where $Q_{\text{KS}}(\cdot)$ represents the survival function of the Kolmogorov distribution:

$$Q_{\text{KS}}(z) = 2 \sum_{k=1}^{\infty} (-1)^{k-1} e^{-2k^2 z^2}.$$

Alternatively, the significance level $\alpha = 0.05$ can be tested using the critical value:

$$D_{0.05} \approx \sqrt{-\frac{1}{2} \ln(0.025)} \cdot \sqrt{\frac{n_1 + n_2}{n_1 n_2}}.$$

If $D > D_{0.05}$, we reject the null hypothesis and consider it as failure (not pass).

**Detailed p-values for VES-Exp and EI Comparison.** We present the p-values obtained from the experiments detailed in Section 4.2. These results are illustrated in Figure 8. It is observed that for the majority of the sample pairs, the calculated p-values are substantially above the $5\%$ significance level.

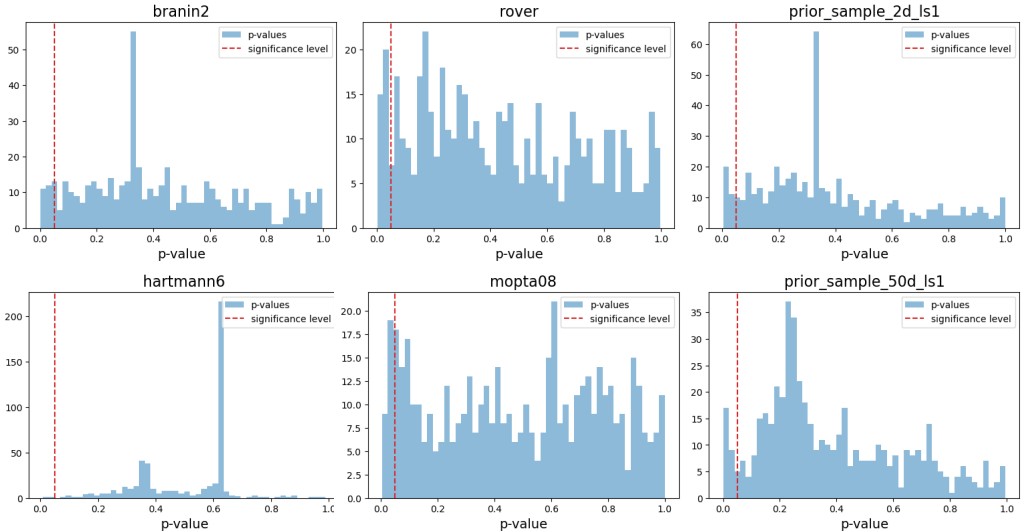

*Figure 8.* Distribution of p-values for 500 sample pairs generated using the EI and VES-Exp acquisition functions.

## D. VES-Gamma Computational Acceleration

Table 2 highlights the higher computational cost of VES methods compared to EI and MES. However, we observe that a technique known as Variable Projection (VarPro) (Golub & Pereyra, 1973; Poon & Peyré, 2023) can be leveraged to accelerate the computation of VES under certain conditions, which VES-Gamma satisfies.

The key idea behind VarPro is that when the function ESLBO has a specific structure,

$$\max_{\boldsymbol{x};k,\beta} \text{ESLBO}(\boldsymbol{x};k,\beta) = \max_{\boldsymbol{x}} \left( \underbrace{\max_{k,\beta} \text{ESLBO}(\boldsymbol{x};k,\beta)}_{\varphi(\boldsymbol{x})} \right), \tag{18}$$

and the solution to $\max_{k,\beta} \text{ESLBO}(\boldsymbol{x};k,\beta)$ is unique, then $\varphi(\boldsymbol{x})$ is differentiable, with

$$\frac{d}{d\boldsymbol{x}} \varphi(\boldsymbol{x}) = \frac{\partial}{\partial \boldsymbol{x}} \text{ESLBO}(\boldsymbol{x}, k_{\boldsymbol{x}}^*, \beta_{\boldsymbol{x}}^*), \tag{19}$$

where $k^*$ and $\beta^*$ are the unique values that maximize ESLBO.

Following the proof in Eq. (13), we establish that the solutions $k_{\boldsymbol{x}}^*$ and $\beta_{\boldsymbol{x}}^*$ are unique. This confirms that it is feasible to implement the VarPro strategy to accelerate the computation of VES-Gamma, eliminating the need for the iterative scheme in Algorithm 1. This ongoing work aims to reduce the computational cost of VES-Gamma to a level comparable to EI and MES.

