# OpenReview forum: "A Unified Framework for Entropy Search and Expected Improvement in Bayesian Optimization"
_ICML.cc/2025/Conference — ICML 2025 oral_

### Official Review · Reviewer_Ru8G · 2025-03-14

**Overall Recommendation:** 4

**Summary:**

The paper derives a novel variational inference scheme for optimizing the Max-value Entropy Search acquisition function in Bayesian optimization. Interestingly, for a particular choice of variational distribution, the classic Expected Improvement acquisition function shows up as a special case, thus showing it to be an approximation of MES, something not previously known. An additional MES approximation, called VES-Gamma, is developed and shows promising performance as an acquisition function.

## update after rebuttal
No change, score already Accept.

**Claims And Evidence:**

The paper does a thorough and rigorous job of supporting its claims that (1) EI is a variational approximation to MES, and (2) we can use the framework to develop other novel acquisition functions that perform well.

**Essential References Not Discussed:**

No.

**Experimental Designs Or Analyses:**

The empirical evaluation of the methods is satisfactory.

**Methods And Evaluation Criteria:**

Yes.

**Other Comments Or Suggestions:**

There is a paragraph in 4.2 describing discrepancies between VES-Exp and EI. Is the EI being used here the LogEI? Earlier in the paper it said LogEI was being used, and used interchangeably with EI. If so that would be another discrepancy potentially worth noting. Would VES-Gamma benefit from a similar strategy to improve optimize-ability?

If there is any commentary on the seemingly larger variance on the Hartmann6 problem that would be interesting to provide.

**Other Strengths And Weaknesses:**

The main strength of the paper is the finding that EI is equivalent to a particular variational approximation to MES. This was not obvious to me and I found it to be very insightful and interesting. I expect other people in the BO community will also be interested to see this.

The further use of the variational approximation framework to derive VES-Gamma was interesting, and I think the fact that it performs only moderately better than MES but at significantly higher computational cost is fine because it was the theoretical insight that is most valuable in the paper.

**Questions For Authors:**

What are the options for improving the runtime?

**Relation To Broader Scientific Literature:**

The paper provides a satisfactory review of acquisition functions and the connections with information-based approaches.

**Theoretical Claims:**

Yes, that EI is a variational approximation for MES using an exponential distribution as the variational distribution.

---

> ### Author Rebuttal · Authors · 2025-03-28
>
> Thank you for your support.
>
> > There is a paragraph in 4.2 describing discrepancies between VES-Exp and EI. Is the EI being used here the LogEI? Earlier in the paper it said LogEI was being used, and used interchangeably with EI. If so that would be another discrepancy potentially worth noting. Would VES-Gamma benefit from a similar strategy to improve optimize-ability?
>
> Thank you for bringing this up. For the comparative analysis with VES-Exp, we used log-EI instead of EI, which could impact their empirical performance differences. We will clarify this in the final draft.
>
> Applying a similar strategy from log-EI to VES-Gamma might be possible, but the exact approach remains unclear. We plan to explore this further in future work.
>
> > If there is any commentary on the seemingly larger variance on the Hartmann6 problem that would be interesting to provide.
>
> We believe this is due to a local optimum of the Hartmann6 function. This was discussed recently in [1].
>
> [1] Battiti, Roberto, and Mauro Brunato. "Pushing the Limits of the Reactive Affine Shaker Algorithm to Higher Dimensions." arXiv preprint arXiv:2502.12877 (2025).
>
> > What are the options for improving the runtime?
>
> To address computational efficiency, we are investigating a \textit{Variable Projection} (VarPro) method from the numerical optimization [2,3]  literature, which allows us to efficiently optimize ESLBO without compromising on performance. VarPro enables this problem reformulation:
>
> $$\max_{\boldsymbol{x}, k, \beta}\text{ESLBO}(\boldsymbol{x}; k, \beta) = \max_{\boldsymbol{x}}\underbrace{\max_{k, \beta}\text{ESLBO}(\boldsymbol{x}; k, \beta)}_{\phi(\boldsymbol{x})},$$
>
> Under the condition that the optimal $k^\ast$ and $\beta^\ast$ exist uniquely, the function $\phi(\boldsymbol{x})$ remains differentiable with
> $$\nabla_{\boldsymbol{x}}\phi(\boldsymbol{x}) = \frac{\partial \text{ESLBO}(\boldsymbol{x}; k^\ast, \beta^\ast)}{\partial \boldsymbol{x}}.$$
> VarPro requires $k^\ast$ and $\beta^\ast$ to be unique. While this is true for the Gamma distribution, it may no longer hold when extended to other distributions.
>
> [2] Golub, G. H. and Pereyra, V. The differentiation of pseudoinverses and nonlinear least squares problems whose variables separate. SIAM Journal on Numerical Analysis, 10(2):413–432, 1973
>
> [3] Poon, C. and Peyré, G. Smooth over-parameterized solvers for non-smooth structured optimization. Mathematical Programming, 201(1):897–952, 2023.

---

> > ### Comment · Reviewer_Ru8G · 2025-04-04
> >
> > Appreciate the answers to my questions.

---

### Official Review · Reviewer_MhFN · 2025-03-14

**Overall Recommendation:** 4

**Summary:**

The paper looks at Bayesian Optimization (BO) and tries to connect two types of acquisition functions that people have always thought of as different approaches. On one side, we have Expected Improvement (EI), which mostly focuses on exploitation by picking points that are likely to be better than what we've already found. On the other side, there are information-theoretic methods like Entropy Search and Max value Entropy Search (MES) that focus more on exploration by reducing uncertainty about where the optimum might be.

What the authors do here is come up with a method called Variational Entropy Search (VES) that shows EI and MES are actually related. They prove that EI is basically a special case of MES when viewed through this variational inference lens. The math involves using the Barber-Agakov bound for mutual information and showing that with a particular choice of variational distribution, you get exactly the EI formula.

They also introduce VES-Gamma, which uses a Gamma distribution as the variational family. This new acquisition function can behave like EI or MES depending on the situation, so it's more flexible.

They tested VES-Gamma on a bunch of problems - toy functions, GP samples, and some real-world optimization tasks. The results show VES-Gamma generally works as well as or better than both EI and MES. It did particularly well on a 388-dimensional SVM hyperparameter tuning problem, and was among the top performers on a 124-dimensional vehicle design problem and a 180-dimensional Lasso feature selection task. So it seems to handle both low and high-dimensional problems pretty well.

**Claims And Evidence:**

The authors claim EI is actually just a special case of MES, viewed through their variational framework. They back this up with math derivations and a theorem that shows when you use an exponential variational distribution, their proposed bound optimization gives the same query point as EI. The math checks out, though it only works for noiseless observations.

They introduce VES-Gamma as a new acquisition function that tries to get the best of both EI and MES. It uses a Gamma distribution variational family (which includes exponential as a special case, so it can recover EI) but adds more flexibility to better match the true maximum value distribution. Their experiments show VES-Gamma does well - often better than both EI and MES across different test problems.

For empirical results, they've got performance curves and statistical tests showing VES-Gamma is at least competitive with, and frequently better than, both EI and MES across their benchmarks. The experimental data supports their claims pretty well. They're honest about limitations though - they note that for the Rover problem, VES-Gamma only performs about as well as EI, not beating MES.

**Essential References Not Discussed:**

The paper covers most of the important papers on EI, MES, and variational methods. They've cited the key works in these areas. But I think they could have mentioned the original EGO paper and maybe Snoek et al. (2012) which talks about practical BO implementation. Adding these wouldn't be essential but would help place their work better in the historical development of Bayesian optimization methods.

**Experimental Designs Or Analyses:**

The experiments are quite thorough, with tests on lots of different benchmarks. This shows the method works well in both simple and complex high-dimensional problems.

As for reproducibility, they've done a good job describing their setup - they include details about GP model settings, how they initialized everything, how many iterations they ran, and they averaged results over multiple runs. All this makes it much easier for others to reproduce their work.

The way they analyze their results seems fair to me. They're upfront about when VES-Gamma does better than other methods and when it just matches them. They also talk about runtime, showing they understand there's a computational tradeoff with their approach.

I like that they used proper statistical tests (like the Kolmogorov-Smirnov test) to compare VES-Exp and EI. This gives more weight to their theoretical claims.

The main downsides I see are the high computational cost of their method and that they only tested in noiseless settings. That said, they do justify these limitations by pointing out that in typical BO applications, function evaluations are usually so expensive that the extra computational overhead isn't a big deal.

**Methods And Evaluation Criteria:**

Proposed Method:
I think the variational inference approach to maximize the lower bound of information gain is pretty novel and makes good sense. They managed to transform that complicated nested expectation in MES into something more tractable by alternating between optimizing the variational distribution and query point. Using the Gamma family for the variational distribution seems reasonable given its flexibility in capturing the true distribution of the maximum value.

Evaluation Benchmarks:
The experiments look good - they cover a nice range of benchmarks from low-dimensional synthetic functions to GP samples and some challenging high-dimensional real-world problems. This gives me confidence that they've tested their method under different conditions.

Metrics and Procedure:
They're using standard metrics (simple regret/best-found value) and their experimental procedures seem appropriate - multiple trials, consistent initialization, and scaling iteration counts with dimensionality. The comparison with EI and MES baselines strengthens their findings.

Comparative Baselines:
While they focus mainly on EI and MES (which makes sense given their paper's goals), it might have been nice to see comparisons with other methods like PES or UCB for more context.

**Other Comments Or Suggestions:**

More intuition about why Gamma vs. Exponential:
The paper should explain better why you chose the Gamma distribution and how its parameters actually balance exploration vs exploitation in practice.

What about VES-Gamma-Sequential?:
You should add a short explanation of the sequential variant in the main paper - it's only in the supplement now, which isn't ideal.

Handling noise?:
The paper only deals with noiseless settings. You should at least discuss how you might extend this to noisy observations, which is a limitation right now.

Future directions:
Have you thought about applying your variational approach to other acquisition functions in BO? Or maybe extending to batch or multi-fidelity settings?

Code release:
Will you release your implementation for reproducibility?

Sensitivity analysis:
I'd like to see some analysis of how sensitive VES-Gamma is to its hyperparameters. This would help practitioners know what to expect.

**Other Strengths And Weaknesses:**

Strengths:
I think the paper has several strong points. First, it's quite original - connecting EI and information-theoretic acquisition functions through this variational framework is a fresh take that nobody's really done before. This is actually a pretty significant conceptual contribution to the field.

The theoretical unification work they did and their VES-Gamma algorithm looks promising for future BO research, especially for those high-dimensional problems that are so challenging. I can see this influencing both theoretical work and practical applications going forward.
The writing is solid - well-structured with clear explanations and they backed everything up with detailed experiments. It's easy to follow their reasoning throughout.

Finally, seeing how well their method performs on those tough high-dimensional tasks makes me think this could be really useful for real-world optimization problems where function evaluations are expensive.

Weaknesses:
There are some limitations worth noting. The computational complexity is a big one - VES-Gamma takes more compute than EI and MES, which might make people hesitant to use it unless they can optimize it further.

I found it limiting that they only tested in noiseless settings. They should extend this to noisy scenarios to make their case stronger.
They claim VES-Gamma dynamically balances exploration and exploitation, but don't really show us how the variational parameters adapt during the optimization process. More insight into this mechanism would help understand what's happening under the hood.

While focusing on EI and MES makes sense, they could have included more baseline comparisons like PES, GP-UCB, or Knowledge Gradient for a more comprehensive evaluation. That would give us a better sense of where this method sits in the broader landscape.

**Questions For Authors:**

I'm curious about how you'd handle noisy settings with VES. Would your EI equivalence still work, or would you need a different approach?

Did you look at how VES-Gamma compares to PES? Do you think your variational approach could work with PES too?

Why Gamma distribution? Did you try other variational families, and how much does performance change if you use something else?

Could you show us more about how the variational parameters actually change during optimization? It would help understand how it balances exploration and exploitation in practice.

For the Rover problem, why did VES-Gamma only match EI when MES did better? Any insights on that?

VES-Gamma seems computationally heavy. Any ideas on making that inner optimization loop faster?

What were the actual numbers from your KS tests comparing VES-Exp and EI? Some p-values or pass rates would be helpful.

Do you think this variational approach could work for other acquisition functions like Knowledge Gradient?

Given the computational cost, what real-world problems would benefit most from VES-Gamma, especially in high dimensions?

**Relation To Broader Scientific Literature:**

The paper does a good job connecting two approaches that have been separate in BO for a while - the EI stuff and the information theory stuff. This unifying view is pretty novel and hasn't really been laid out clearly before.

The authors aren't working in a vacuum here - they're clearly building on previous research like Entropy Search, Predictive Entropy Search, and MES, and they bring in ideas from variational inference. Their work fits well within the field and they cite all the important papers in BO you'd expect.

They mostly focus on EI and MES in their comparisons, but they do mention other methods like GP-UCB and Knowledge Gradient in the related work section. What's nice is that instead of just heuristically combining different acquisition functions like some other work, their framework gives a more principled way to unify these approaches.

**Theoretical Claims:**

I think the derivation of the variational lower bound on the MES objective looks solid - they used the Barber-Agakov bound which makes sense here. This gives them a good theoretical foundation for connecting information-based acquisition functions to variational inference approaches.

Their main theoretical result (Theorem 3.2) shows that when you use an exponential variational distribution, their method gives you exactly the same thing as EI. The math checks out, and they even verified this empirically with a Kolmogorov-Smirnov test, which is nice. Note that this only works in the noiseless setting though.

For the VES-Gamma part, they don't give us a formal theorem, but the intuition makes sense - moving from exponential to Gamma distribution lets the method capture more complex behaviors in the true distribution. The experiments back this up, so I'm convinced even without a formal proof here.

---

> ### Author Rebuttal · Authors · 2025-03-29
>
> We sincerely thank you for the detailed comments and support on this work. Due to word limit constraints we will give brief answers:
>
> > High computational cost
>
> We are investigating the VarPro method, as detailed in our response to reviewer VbQ5.
>
> > EGO and Snoek et al.
>
> We will include them in the Related Work section.
>
> > How the variational parameters adapt
>
> We agree that visualizing the evolution of $k$ and $\beta$ would provide valuable insight, but it's challenging since their values depend on the query point $x$, which changes at each iteration. Simply showing their values at each step wouldn't effectively capture the trends.
>
> We have put considerable effort into understanding VES-Gamma's mechanics and tried to formalize our findings into a theorem. However, we acknowledge the difficulty of doing so rigorously and will continue exploring this in future work.
>
> > More baseline
>
> Due to rebuttal time constraints, we focus on generating results using synthetic functions for UCB-0.1 and KG: https://ibb.co/4n0Nxpnv. We met numerical issues running PES and are working on fixing it.
>
> >  Why Gamma distribution
>
> We chose the Gamma distribution because it is a generalization of the exponential distribution and it is characterized by a limited degree of freedom. We also tested the generalized Gamma distribution, but its extra flexibility prevents solving for $k$ and $\beta$ in closed form.
>
> Regarding the impact of $k$ and $\beta$ on exploration vs. exploitation, we hypothesize that larger $k$ promotes exploration by weakening the EI term. However, we lack strong numerical or theoretical evidence, suggesting that there is no simple relationship.
>
> > VES-Gamma-Sequential
>
> Apologies for the confusion. We initially included VES-Gamma-Sequential as a potential extension of VES-Gamma but later decided to remove it due to unresolved issues. Some references were mistakenly left in the appendix, which we will remove in the final version.
>
> > Handling noise
>
> The noiseless assumption is critical to our theoretical analysis, as detailed in our response to reviewer VbQ5.
>
> > Future direction of variational approach
>
> Extending the current work to include other acquisition functions and exploring the noisy setting, batch, multi-fidelity, and multi-objective optimization are promising future research directions.
>
> > Code release
>
> Yes
>
> > Sensitivity analysis
>
> We conducted ablation studies on two key parameters: the clamping threshold for $z_{\boldsymbol{x}}^\ast$ and the number of samples used to estimate the expectation. The results can be found at https://ibb.co/RpY4MXbQ and https://ibb.co/v4dsvNh4, respectively. Each experiment was repeated 10 times to compute the uncertainty bars. The findings indicate that VES-Gamma is relatively robust to variations in these parameters.
>
> > EI and VES-Exp equivalence with noise
>
> Unfortunately, in noisy settings, the current theory breaks down. Alternative approaches will be necessary to address this issue.
>
> > VES to PES
>
> The challenge is selecting a flexible variational family that captures the characteristics of $x^\ast$ while remaining computationally feasible. Due to the multi-modality of $x^\ast$, a Gaussian or Beta mixture may be needed, but handling the ESLBO with a mixture density is complex. This poses a major challenge in extending VES to PES, though we remain optimistic about future research on this!
>
> > Rover problem
>
> While it is difficult to draw a definitive conclusion, we observe that all three methods fall within the uncertainty bounds. We will conduct additional repetitions to assess whether MES's superiority is statistically significant.
>
> > KS tests p-values
>
> More p-value details of benchmarks are available: https://ibb.co/WpkPJqWq. This information will also be included in the appendix of the final version.
>
> > VES for KG
>
> It is possible. The main challenge lies in designing a suitable variational family for the Knowledge Gradient acquisition function.
>
> > What problems for VES
>
> We observe that VES-Gamma performs better in high-dimensional problems, and we believe it is well-suited for solving such problems with expensive function evaluations.

---

### Official Review · Reviewer_VbQ5 · 2025-03-16

**Overall Recommendation:** 4

**Summary:**

This paper introduces "Variational Entropy Search" (VES), a unified theoretical framework that reveals a previously unrecognized connection between Expected Improvement (EI) and information-theoretic acquisition functions in Bayesian optimization. The authors demonstrate that EI, traditionally considered distinct from information-theoretic methods, can be interpreted as a variational lower bound on the Max-value Entropy Search (MES) acquisition function when using an exponential distribution as the variational distribution.

Building on this theoretical insight, the authors propose VES-Gamma, a novel acquisition function that employs a Gamma distribution (which generalizes the exponential distribution) to approximate the maximum value distribution. This approach creates an intermediary between EI and MES that balances their respective strengths. The ESLBO (Entropy Search Lower Bound) objective in VES-Gamma includes EI as one of its components, with additional terms that dynamically adjust the exploration-exploitation trade-off.

Through empirical evaluations across low and high-dimensional synthetic benchmarks, GP samples, and real-world problems, the authors claim that VES-Gamma performs competitively with and can even outperform both EI and MES.

**Claims And Evidence:**

The paper's claims appear to be supported by both theoretical derivations and empirical evidence.

**Essential References Not Discussed:**

No essential references appear to be missing

**Experimental Designs Or Analyses:**

The paper employs a sound experimental methodology, evaluating algorithms across diverse benchmarks that span synthetic functions (Branin, Levy, Hartmann, Griewank), GP samples with varying dimensionality (2D to 100D), and complex real-world problems (Rover, Mopta08, Lasso-DNA, SVM) with dimensions up to 388D. The experimental protocol follows good practices including: consistent GP hyperparameter settings across all methods, proper warm-starting with 20 random samples, repeated trials (10 per experiment) with statistical reporting, and appropriate metrics (simple regret). The authors also carried out a Kolmogorov-Smirnov test validation of the theoretical equivalence between VES-Exp and EI that showed high passing rates (>90%) across all benchmarks.

A few concerns worth highlighting: the substantial computational cost disparity between VES (53.17s per iteration) versus EI/MES (~1-1.6s) limits practical applicability; the timeout issues on very high-dimensional problems (e.g., 1000D Lasso-Hard); and the restriction to noiseless settings. However, these limitations are transparently acknowledged by the authors, and the overall experimental design remains sound and supports the paper's claims.

**Methods And Evaluation Criteria:**

The methods and evaluation criteria in this paper are well-aligned with its theoretical contributions and practical goals. The VES framework employs mathematically sound variational methods to establish the connection between EI and MES, while VES-Gamma represents a natural theoretical extension. The evaluation approach is appropriate, using simple regret as the performance metric and directly comparing against the most relevant baselines (EI and MES). The benchmark selection is reasonably comprehensive, spanning synthetic functions (Branin, Levy, Hartmann, Griewank), GP samples with varying length scales, and real-world problems (Rover, Mopta08, Lasso-DNA, SVM) across dimensions from 2D to 388D. This diverse test suite provides sound evidence for both the theoretical claims and practical performance benefits of the proposed approach.

**Other Comments Or Suggestions:**

-

**Other Strengths And Weaknesses:**

A notable weakness is the significant computational overhead of VES-Gamma (53.17s per iteration versus 1.63s for EI), which could limit its practical adoption despite performance gains. Additionally, the framework currently only addresses noiseless settings, an important limitation that the authors acknowledge needs addressing in future work.

**Questions For Authors:**

-

**Relation To Broader Scientific Literature:**

The paper bridges a conceptual gap in Bayesian optimization by revealing that Expected Improvement and information-theoretic approaches like Max-value Entropy Search are variations of the same underlying framework rather than distinct methodologies. This theoretical unification challenges conventional wisdom in the field, where these approaches have been treated as separate paradigms, while also delivering a practical payoff through VES-Gamma, which leverages this insight to balance exploration and exploitation more effectively than either approach alone.

**Theoretical Claims:**

I examined the two primary theoretical claims in the paper: Theorem 3.1 (ESLB Proof) and Theorem 3.2 (VES-Exp and EI Equivalence). Both proofs appear mathematically sound, though I note that this theoretical equivalence depends on the noiseless observation assumption, which the authors acknowledge as a limitation. The proof legitimately establishes the connection, though its practical applicability has the constraints noted by the authors. The mathematical development of both theorems appears correct and follows established techniques from variational inference.

---

> ### Author Rebuttal · Authors · 2025-03-28
>
> We sincerely appreciate your support for this work.
> > the substantial computational cost disparity between VES (53.17s per iteration) versus EI/MES (~1-1.6s) limits practical applicability; the timeout issues on very high-dimensional problems (e.g., 1000D Lasso-Hard);
>
> To address computational efficiency, we are investigating a *Variable Projection* (VarPro) method from the numerical optimization [1,2]  literature, which allows us to efficiently optimize ESLBO without compromising on performance. VarPro enables this problem reformulation:
>
> $$\max_{\boldsymbol{x}, k, \beta}\text{ESLBO}(\boldsymbol{x}; k, \beta) = \max_{\boldsymbol{x}}\underbrace{\max_{k, \beta}\text{ESLBO}(\boldsymbol{x}; k, \beta)}_{\phi(\boldsymbol{x})},$$
>
> Under the condition that the optimal $k^\ast$ and $\beta^\ast$ exist uniquely, the function $\phi(\boldsymbol{x})$ remains differentiable with
> $$\nabla_{\boldsymbol{x}}\phi(\boldsymbol{x}) = \frac{\partial \text{ESLBO}(\boldsymbol{x}; k^\ast, \beta^\ast)}{\partial \boldsymbol{x}}.$$
> VarPro requires $k^\ast$ and $\beta^\ast$ to be unique. While this is true for the Gamma distribution, it may no longer hold when extended to other distributions.
>
> [1] Golub, G. H. and Pereyra, V. The differentiation of pseudoinverses and nonlinear least squares problems whose variables separate. SIAM Journal on Numerical Analysis, 10(2):413–432, 1973
>
> [2] Poon, C. and Peyré, G. Smooth over-parameterized solvers for non-smooth structured optimization. Mathematical Programming, 201(1):897–952, 2023.
>
> > and the restriction to noiseless settings
>
> We recognize that observation noise plays a crucial role in Bayesian optimization. The primary reason for assuming noiseless observations in this work is that our theoretical framework relies on this assumption. Specifically, we assume that the support of $p(y^\ast \mid D_t, y_x)$ is $[\max\{y_x, y^\ast_t\}, +\infty)$, which may no longer hold if $y_x$ is noisy.
>
> We also note that EI and MES, the acquisition functions most closely related to VES, were derived under a noise-free assumption. Furthermore, many real-world problems really are noiseless, including the benchmarks considered in our paper. Adapting VES to handle observation noise is a promising avenue for future research, and we plan to explore this direction in subsequent work.

---

### Decision · Program_Chairs · 2025-05-01

**Decision:**

Accept (oral)

**Comment:**

This work develops a variational inference approach to approximating a lower bound of Max-value Entropy Search (MES).  The authors demonstrate a novel connection between the classic EI acquisition function and MES (under the exponential variational distribution).  Reviews all found the work to be novel, insightful, and theoretically sound.  The authors demonstrate the performance of the acquisition function empirically through simulation studies on a variety of test problems of varying dimensionality.  The computational cost of the AF (and perhaps, ambiguous performance relative to other SoTA methods like Joint-Entropy Search) limit its practical use, the contribution remains clear.

The reviewers raised a number of points that are important to discuss in the CR, including discussion of limitations wrt noisy observations.  As discussed in the paper and in the reviews, the work would be more compelling if the computational cost were lower—the authors appear to have identified a promising numerical approach which I hope they are able to leverage.  Finally, additional comparisons with other AFs and perhaps the addition of a few more test problems can help readers understand where the empirical performance stacks up.